# In Systemic Sclerosis Patients, Peripheral Blood CD21^low^ B Cells and Serum IL-4 and IL-21 Influence Joint Involvement

**DOI:** 10.3390/jpm13091334

**Published:** 2023-08-30

**Authors:** Chiara Pellicano, Amalia Colalillo, Valeria Carnazzo, Serena Redi, Valerio Basile, Mariapaola Marino, Umberto Basile, Edoardo Rosato

**Affiliations:** 1Department of Translational and Precision Medicine, Sapienza University of Rome, 00185 Rome, Italy; chiara.pellicano@uniroma1.it (C.P.); amalia.colalillo@uniroma1.it (A.C.); edoardo.rosato@uniroma1.it (E.R.); 2UOC of Clinical Pathology DEA II Level, Hospital Santa Maria Goretti-ASL Latina, 04100 Latina, Italy; v.carnazzo@ausl.latina.it (V.C.);; 3Clinical Pathology and Cancer Biobank IRCCS Regina Elena National Cancer Institute, 00128 Rome, Italy; 4Dipartimento di Medicina e Chirurgia Traslazionale, Sezione di Patologia Generale, Università Cattolica del Sacro Cuore, Fondazione Policlinico Universitario A. Gemelli, 00168 Rome, Italy; mariapaola.marino@unicatt.it

**Keywords:** systemic sclerosis, arthritis, joint, DAS28, IL-4, IL-21, CD21^low^

## Abstract

Systemic sclerosis (SSc) patients have an increased frequency of CD21^low^ B cells and of serum interleukin-4 (IL-4) and IL-21, each possible markers of joint involvement in inflammatory arthritis. The aim of this study was to investigate the possible influence of CD21^low^ B cells, IL-4, and IL-21 on joint involvement in a cohort of 52 SSc patients. The DAS28-ESR was correlated with CD21^low^ B cells (r = 0.452, *p* < 0.001), IL-4 (r = 0.478, *p* < 0.001), and IL-21 (r = 0.415, *p* < 0.001). SSc patients with a DAS28-ESR > 3.2 had more CD21^low^ B cells (12.65% (IQR: 7.11–13.79) vs. 5.08% (IQR: 3.76–7.45), *p* < 0.01), higher IL-4 levels (132.98 pg/mL (IQR: 99.12–164.12) vs. 100.80 pg/mL (IQR: 62.78–121.13), *p* < 0.05), and higher IL-21 levels (200.77 pg/mL (IQR: 130.13–302.41) vs. 98.83 pg/mL (IQR: 35.70–231.55), *p* < 0.01) than patients with a DAS28-ESR ≤ 3.2. The logistic regression analysis models showed that the DAI (OR: 2.158 (95% CI: 1.120; 4.156), *p* < 0.05) and CD21^low^ B cells (OR: 1.301 (95% CI: 1.099; 1.540), *p* < 0.01), the DAI (OR: 2.060 (95% CI: 1.082; 3.919), *p* < 0.05) and IL-4 level (OR: 1.026 (95% CI: 1.006; 1.045), *p* < 0.01), and the DAI (OR: 1.743 (95% CI: 1.022; 2.975), *p* < 0.05) and IL-21 level (OR: 1.006 (95% CI: 1.000; 1.011), *p* < 0.05) were independently associated with a DAS28-ESR > 3.2. An elevated CD21^low^ B cell percentage, IL-4 level, and IL-21 level was associated with higher articular disease activity in patients, suggesting a possible role in the pathogenesis of SSc joint involvement.

## 1. Introduction

Systemic sclerosis (SSc) is an autoimmune disease characterized by microvascular abnormalities due to endothelial dysfunction, skin and internal organ fibrosis, and aberrant immune system activation with specific autoantibody production [1]. Articular manifestations, such as generalized arthralgias, arthritis, joint contractures, and tendon friction rubs (TFRs), are common in SSc, involving 46–97% of SSc patients during the disease course, but mainly affecting patients with early diffuse SSc [2]. Joint involvement in SSc patients is significantly associated with a reduced quality of life and disabilities [3], and mainly affects the small joints of the hand, such as the metacarpophalangeal (MCP) and proximal interphalangeal (PIP) joints, wrists, and ankles [4]. Although the disease activity score of 28 joints (DAS28) is a validated score for assessing disease activity in rheumatoid arthritis (RA) patients, it is also used to assess joint involvement in SSc patients [5,6,7]. It is a composite index, including the number of tender and/or swollen joints, the general health, and an inflammatory marker (C-reactive protein (CRP) for the DAS28-CRP or erythrocyte sedimentation rate (ESR) for the DAS28-ESR), that allows the degree of joint inflammation activity to be evaluated [6,7]. The DAS28-ESR showed the best performance regarding reliability and construct validity for assessing arthritis in SSc patients [6]. The pathogenesis of SSc is still partially unknown, but several studies have confirmed a critical role of peripheral blood B cells in systemic autoimmunity and disease expression due to their capacity to produce several inflammatory and profibrotic cytokines [8]. Diffuse infiltrate or small focal aggregates of lymphocytes and plasma cells have been reported in synovial biopsies of the joints of SSc patients [9]. Moreover, in SSc patients, there is an increased frequency of CD21^low^ B cells [10]. This population of anergic and exhausted B cells, due to chronic immune stimulation, could play the role of antigen-presenting cells (APCs) and possibly accumulate in inflamed tissues [10]. We previously demonstrated that SSc patients had more stable-over-time peripheral blood CD21^low^ B cells compared to healthy controls (HCs), and peripheral blood CD21^low^ B cells are associated with visceral vascular complications and impaired angiogenesis in SSc patients [10,11]. An increased percentage of peripheral blood and synovial fluid CD21^low^ B cells was also found in RA patients, and the percentage of this subset was correlated with joint erosions and destruction [12]. Moreover, a high percentage of CD21^low^ B cells was found in patients with axial spondylarthritis (axSpA) [13] and juvenile idiopathic arthritis (JIA) [14]. Interestingly, in patients with JIA, the synovial fluid CD21^low^ B cells were correlated with peripheral T helper (pTh) cells, which, by provision of interleukin 21 (IL-21), skewed B cell differentiation toward a CD21^low^ phenotype in vitro [15]. Several studies have provided experimental evidence elucidating the multifaceted role of IL-21 in RA disease progression, due to its role in activating T cells, B cells, monocytes/macrophages, and synovial fibroblasts [16]. IL-4 has recently emerged as a potential player in the pathogenesis of inflammatory arthritis, since enhanced IL-4 concentrations were found in the synovial fluid and plasma samples of RA patients, and were also found before disease development [17]. SSc patients had higher serum levels of both IL-4 and IL-21 compared to HCs, and the serum levels of these cytokines were correlated with a reduction in the diffusion lung capacity for carbon monoxide (DLco) and the radiological extension of interstitial lung disease (ILD) in SSc patients [18]. The aim of this study was to investigate the possible influence of peripheral blood CD21^low^ B cells and serum IL-4 and IL-21 levels on joint involvement in a cohort of SSc patients.

## 2. Materials and Methods

### 2.1. Subjects

Fifty-two consecutive SSc patients were enrolled in this study. All patients fulfilled the American College of Rheumatology/European League Against Rheumatism collaborative criteria (2013 ACR/EULAR) for SSc [19]. The exclusion criteria were: an age < 18 years or >70 years, cardiopulmonary diseases not related to SSc, pulmonary arterial hypertension (PAH), a history of infection in the last 3 months, malignancies, osteoarthritis, microcrystalline or infectious arthritis, allergic diseases, other autoimmune diseases, and positivity for the rheumatoid factor (RF) or for anti-citrullinated protein antibodies (ACPAs). We also excluded smokers and pregnant or breastfeeding women from the study. Finally, SSc patients treated in the last 6 months with immunosuppressive agents, such as methotrexate (MTX), mycophenolate mofetil (MMF), TNF-a inhibitors, or rituximab (RTX), and corticosteroids at an equivalent dose of prednisone ≥ 10 mg/day were excluded.

Written informed consent was obtained from all the subjects enrolled in the study. The study was conducted according to the Declaration of Helsinki. The ethics committee of the Sapienza University of Rome approved this study (protocol number 0304).

### 2.2. Clinical Assessment

The modified Rodnan skin score (mRSS) was used to assess skin involvement and the disease subset (diffuse cutaneous SSc (dcSSc) or limited cutaneous SSc (lcSSc)) was defined according to LeRoy et al. [20]. The disease activity index (DAI) was measured as follows: Δ-skin = 1.5, mRSS > 18 = 1.5, digital ulcers (DUs) = 1.5, TFRs = 2.25, CRP > 1 mg/dL = 2.25, and DLco % predicted <70% = 1.0; a cut-off of ≥2.5 identified patients with an active disease [21]. The disease severity scale (DSS) evaluates the involvement of the general state, peripheral vessels, skin, joints/tendons, muscles, gastrointestinal tract, lungs, heart, and kidneys. Each organ or system is assessed separately with a score ranging from a minimum of 0 to a maximum of 4, representing its null, intermediate, moderate, severe, or end-stage involvement; the higher the score, the greater the disease severity [22]. The inflammatory articular involvement was assessed using the DAS28-ESR, and a value of ≤3.2 was defined as the threshold for a low disease activity state [23]. Nailfold videocapillaroscopy (NVC) was performed to evaluate the morphology of the nailfold dermal papillary capillaries using a videocapillaroscope equipped with a 500× magnification lens (Pinnacle Studio version 8 software, Corel, Ottawa, ON, Canada). The capillaroscopic images were classified according to the early, active, and late patterns according to Cutolo et al. [24].

### 2.3. Laboratory Assessment

The immunophenotyping of peripheral blood samples was performed using combinations of fluorochrome-labeled monoclonal antibodies (Becton-Dickinson Biosciences) to CD19-PC5.5, CD21-PE, CD27-APC, and IgD-FITC. A flow cytometry analysis was performed on a BD FACSCalibur system, and the data files were acquired and analyzed using the CELLQuest 3.3 (Becton Dickinson, Mountain View, CA, USA) and FlowJo (TreeStar, Ashland, Ore, OR, USA) version 10.7.1 software. Peripheral venous blood samples were collected in tubes and centrifuged at 3000× *g* for 15 min at 19 °C. Serum samples were aliquoted into 1.5 mL Eppendorf tubes and stored at −80 °C until the time of the assay. Specimens were thawed only once, and were immediately assayed in a blinded fashion and in a single batch. The assessment of the serum levels of IL-4 and IL-21 cytokines was carried out using Bio-Plex Multiplex Immunoassays, which allow the simultaneous measurement of analytes and are available as ready-to-use premixed multiplex panels (BIO-RAD, Hercules, CA, USA). Antinuclear antibodies (ANAs), extractable nuclear antigens (ENAs), RF, ACPAs, CRP, and the ESR were also evaluated. The antibody profile was evaluated using indirect immunofluorescence (IFI) for the detection of ANAs, and the definition of the immunofluorescence pattern (homogeneous, speckled, centromeric, or cytoplasmic) was carried out according to the international consensus on ANA patterns [25]. Specific commercial ELISA kits were used for the detection of ENAs, RF, ACPAs, and CRP. An automated method was used for the ESR assessment. All blood tests were performed in a single analytical session by an expert operator without knowledge about the clinical information of the handled sample. Each sample was tested twice to minimize eventual discrepancies, and all tests were performed in the same laboratory with the same instruments. Moreover, samples with serum dilutions, when necessary, were tested according to the manufacturer’s instructions and recommendations. Potentially harmful samples were handled by taking all safety precautions.

### 2.4. Statistical Analysis

The SPSS version 26.0 software (Bioz, Los Altos, CA, USA) was used for the statistical analyses. The normal distribution of data was evaluated using the Shapiro–Wilk test. Continuous variables were expressed as the mean and standard deviation (SD) or median and interquartile range (IQR) as appropriate. Categorical variables were expressed as absolute frequencies and percentages. Student’s *t*-test or the Mann–Whitney U test was performed to evaluate the differences between groups. Differences between categorical variables were evaluated using the chi-squared or Fisher’s exact test as appropriate. The Pearson or Spearman correlation tests were used for bivariate correlations as appropriate. Multivariate logistic regression models with the odds ratio (OR) and a 95% confidence interval (CI) were applied to analyze the CD21^low^ B cell percentage, IL-4 level, and IL-21 level for the DAS28-ESR. A *p*-value < 0.05 was considered significant.

## 3. Results

### 3.1. Demographic and Clinical Characteristics of Enrolled SSc Patients

The 52 enrolled SSc patients had a median age of 57.5 years (IQR: 48.75–63) with a median disease duration of 11.5 years (IQR: 6–16). The SSc patients were predominantly female (92.3% vs. 7.7%). A total of 29 (55.8%) SSc patients had lcSSc and 23 (44.2%) had dcSSc, with a median mRSS of 11 (IQR: 7.75–15.25). An early NVC pattern was present in 10 (19.3%) patients whilst 15 (28.8%) patients had an active NVC pattern and 27 (51.9%) patients had a late NVC pattern. Scl70 or anti-RNA polymerase III positivity was found in 25 (48.1%) patients and 2 (3.8%) patients, respectively, whilst 12 (23.1%) and 13 (25%) patients had an anti-centromere or only ANA positivity, respectively. The median DAI and DSS were 2.42 (IQR: 1.26–4) and 7 (IQR: 6–9), respectively. The reported median visual analog scale (VAS) for arthritis was 2.5 (IQR: 2–5.25). The median DAS28-ESR was 2.32 (IQR: 1.5–3.57), and 16 (30.77%) patients had a DAS28-ESR > 3.2. The median percentage of peripheral blood CD21^low^ B cells was 6.3% of the total B cells (IQR: 3.97–13), whilst the median serum IL-4 level was 110.22 pg/mL (IQR: 87.03–140.8) and the median serum IL-21 level was 130.18 pg/mL (IQR: 53.93–255.02). Table 1 summarizes the demographic and clinical characteristics of the SSc patients enrolled in this study.

### 3.2. Peripheral Blood CD21^low^ B Cells and Serum IL-4 and IL-21 Levels

We found a slightly significant positive linear correlation between the percentage of peripheral blood CD21^low^ B cells and the serum IL-4 level (r = 0.315; *p* < 0.05) (Figure 1a) and serum IL-21 level (r = 0.351; *p* < 0.01) (Figure 1b). Moreover, a significant positive linear correlation was found between the serum IL-4 and serum IL-21 levels (r = 0.468; *p* < 0.001) (Figure 1c).

### 3.3. Joint Involvement, Peripheral Blood CD21^low^ B Cells, and Serum IL-4 and IL-21 Levels

We found a slightly significant positive linear correlation between the VAS for arthritis and peripheral blood CD21^low^ B cells (r = 0.361; *p* < 0.01) (Figure 2a). Statistically significant positive linear correlations were found between the VAS for arthritis and both the serum IL-4 level (r = 0.470; *p* < 0.001) (Figure 2b) and the serum IL-21 level (r = 0.403; *p* < 0.01) (Figure 2c). Moreover, we found statistically significant positive linear correlations between the DAS28-ESR and peripheral blood CD21^low^ B cells (r = 0.452; *p* < 0.001) (Figure 2d), the serum IL-4 level (r = 0.478; *p* < 0.001) (Figure 2e), and the serum IL-21 level (r = 0.415; *p* < 0.001) (Figure 2f).

SSc patients with a DAS28-ESR > 3.2 had a statistically significantly higher percentage of peripheral blood CD21^low^ B cells (12.65% of total B cells (IQR: 7.11–13.79) vs. 5.08% of total B cells (IQR: 3.76–7.45), *p* < 0.01) (Figure 3a), a higher serum IL-4 level (132.98 pg/mL (IQR: 99.12–164.12) vs. 100.80 pg/mL (IQR: 62.78–121.13), *p* < 0.05) (Figure 3b), and a higher serum IL-21 level (200.77 pg/mL (IQR: 130.13–302.41) vs. 98.83 pg/mL (IQR: 35.70–231.55), *p* < 0.01) (Figure 3c) compared to SSc patients with a DAS28-ESR ≤ 3.2.

We constructed three models of a multivariable logistic regression analysis to evaluate the association between a DAS28-ESR > 3.2 and the independent variables. In the first model, the DAI (OR: 2.158 (95% CI: 1.120; 4.156), *p* < 0.05) and the percentage of peripheral blood CD21^low^ B cells (OR: 1.301 (95% CI: 1.099; 1.540), *p* < 0.01) were independently associated with a DAS28-ESR > 3.2. The second model showed that the DAI (OR: 2.060 (95% CI: 1.082; 3.919), *p* < 0.05) and the serum IL-4 level (OR: 1.026 (95% CI: 1.006; 1.045), *p* < 0.01) were independently associated with a DAS28-ESR > 3.2. Finally, in the third model, the DAI (OR: 1.743 (95% CI: 1.022; 2.975), *p* < 0.05) and the IL-21 level (OR: 1.006 (95% CI: 1.000; 1.011), *p* < 0.05) were independently associated with a DAS28-ESR > 3.2. Table 2 shows all the models of the multivariable logistic regression analysis.

## 4. Discussion

In this study, we found an increased percentage of peripheral blood CD21^low^ B cells and increased serum IL-4 and IL-21 levels in SSc patients with joint involvement. The peripheral blood CD21^low^ B cell percentage and the serum IL-4 and IL-21 levels were higher in SSc patients with a DAS28-ESR > 3.2. Moreover, the peripheral blood CD21^low^ B cell percentage and serum IL-4 and IL-21 levels were positively correlated with the DAS28-ESR and VAS for arthritis. Finally, the peripheral blood CD21^low^ B cell percentage and the serum IL-4 and IL-21 levels were independently associated with a DAS28-ESR > 3.2.

Joint involvement in SSc patients as an early manifestation of the disease, often also before the occurrence of the Raynaud’s phenomenon, ranges from 12% to 65% of SSc patients [2]. Usually, the articular involvement in SSc patients encompasses generalized arthralgias with slight pain and stiffness and an oligoarticular or polyarticular pattern, and manifestations tend to be intermittent or chronic-remittent [2]. In the literature, the estimated prevalence of arthritis detected by a physical examination in unselected SSc patients ranges from 7% to 60% [6]. In our study, 16 (30.77%) SSc patients had a DAS28-ESR > 3.2, indicating an active inflammatory articular disease, and, in line with the data from a recent report [6], the median DAS28-ESR was 2.32 (IQR: 1.5–3.57). Joint involvement is one of the primary causes of disabilities and a poor quality of life in SSc patients, so the identification of the underlying pathophysiologic mechanisms is crucial for identifying new potential therapeutic targets.

SSc is a complex autoimmune disease that shares several clinical and immunological features with RA, such as common susceptibility loci and the dysregulation of innate and adaptive immunity, including the activation of self-reactive B cells able to secrete autoantibodies and systemic complications [26]. The key role of B cells in the pathogenesis of SSc, which is characterized by a low-grade inflammatory response with respect to other autoimmune diseases, is demonstrated by the production of immunoglobulin free light chains in the serum (sFLCs) that reflect the clone expansion of B cells [27]. Moreover, the chronic immune stimulation in SSc patients is responsible for the accumulation of CD21^low^ B cells, a particular B cell population of functionally anergic and exhausted cells that express a high level of activation markers, inhibitory receptors, and a peculiar pattern of homing receptors [10]. The exact role of CD21^low^ B cells is still poorly understood; however, chronic antigen stimulation provided by infections or by autoantigens might favor their accumulation in peripheral blood and in peripheral inflammatory tissue [28]. It has been demonstrated that, in RA patients, CD21^low^ B cells constituted almost half of the B cells in the synovial fluid of the inflamed joints, due to their expression of the inflammatory chemokine receptor CXCR-3 and the secretion of its ligand, CXCL-9 [12]. Moreover, in RA patients, CD21^low^ B cells might have a role in joint damage via bone erosion by secreting the receptor activator of nuclear factor kappa-B ligand (RANKL) and IL-6 [12]. An increase in circulating CD21^low^ B cells was also found in axSpA patients, and in this cohort of patients, CD21^low^ B cells were correlated with the ESR [13]. Dirks et al. [14] demonstrated an accumulation of CD21^low^ B cells in the synovial fluid of ANA + JIA patients, potentially triggered by the (auto)antigens present at the site of inflammation.

We previously demonstrated a stable-over-time expansion of peripheral blood CD21^low^ B cells in SSc patients [10,11]. Moreover, in SSc patients, an increased percentage of CD21^low^ B cells was associated with visceral vascular complications and impaired angiogenesis, suggesting a potential role in the pathogenesis and progression of the disease [10,11].

To the best of our knowledge, no study to date has specifically investigated the role of CD21^low^ B cells in SSc articular involvement.

In this study, we demonstrated positive linear correlations between peripheral blood CD21^low^ B cells and both the VAS for arthritis and the DAS28-ESR in SSc patients, suggesting a possible role of this particular subset of the B cell population in the pathogenesis of joint involvement. Moreover, we demonstrated that SSc patients with active articular involvement, as demonstrated by a DAS28-ESR > 3.2, had a statistically significantly higher percentage of peripheral blood CD21^low^ B cells compared to SSc patients without joint involvement. This result is strengthened by the multivariable logistic regression analysis, which showed that peripheral blood CD21^low^ B cells were independently associated with a DAS28-ESR > 3.2. These findings and the peculiar characteristic of peripheral blood CD21^low^ B cells potentially migrating towards sites of inflammation might be of particular relevance in the pathogenesis of SSc articular involvement and in the progression of the disease. In further studies, it would be interesting to evaluate the possible influence of immunosuppressive agents (i.e., methotrexate, mycophenolate mofetil, TNF-a inhibitors, or rituximab) and corticosteroids at an equivalent dose of prednisone ≥ 10 mg/day on the peripheral blood CD21^low^ B cells and on the DAS28-ESR assessment, since, in this study, we excluded SSc patients that received these treatments in the last 6 months.

The differentiation of CD21^low^ B cells depends on the interaction with CD4^+^ T helper (Th) cells and their secreted cytokines [29]. Extra-follicular Th cells could be detected in the inflamed tissues of patients with autoimmune diseases as well as murine disease models [30,31]. This particular T cell subset accumulates in the joints of RA patients and of ANA + JIA patients, where they secrete a large amount of IL-21 [15]. In ANA + JIA patients, it has been recently demonstrated that this subset of Th cells possesses a potent “B-helper function”, not only inducing plasma cell differentiation, but also favoring B-cell differentiation towards a CD21^low^ B cell phenotype through the influence of IL-21 and IFN-γ [15].

IL-21 is a Th2 cytokine with multifaceted roles in activating T cells, B cells, monocytes/macrophages, and synovial fibroblasts in the RA pathogenesis through the activation of the JAK/STAT, MAPK, and PI3K/Akt signaling pathways [16]. IL-21 is secreted by several cell types, such as follicular Th, Th17, and natural killer (NK) cells, and it has a key role in the preservation of the germinal center, in polarization through Th17, and in antibody switching [32]. In SSc patients, an increased serum level of IL-21 compared to HCs and a correlation between the serum IL-21 levels and the fibrotic involvement of the lungs have been demonstrated [18]. Moreover, an overexpression of IL-21 mRNA has been found in the keratinocytes of SSc patients compared to HCs [33]. It has been demonstrated that IL-21 may induce fibrosis favoring the Th2 polarization, since a link between IL-21 mRNA and IL-4 mRNA exists [33]. Ly et al. [34] showed a correlation between antibody production and inflammation in SSc patients and, more interestingly, an imbalance in follicular Th with a greater capacity to secrete IL-21 and IL-4.

In this study, we found a positive linear correlation between serum IL-21 levels and the percentage of peripheral blood CD21^low^ B cells, confirming data in the literature about the link between this cytokine and the expansion of this B cell subset. Moreover, as expected, we found a correlation between serum IL-21 and serum IL-4 levels. More interestingly, we showed a positive linear correlation between the serum IL-21 levels and the VAS for arthritis or the DAS-28-ESR, and SSc patients with moderate–high joint involvement had statistically significantly higher serum IL-21 levels than SSc patients with a DAS28-ESR ≤ 3.2. This correlation probably reflects the correlation between peripheral blood CD21^low^ B cells and articular involvement in SSc patients. Finally, in a multivariable logistic regression analysis, we demonstrated that serum IL-21 levels were independently associated with a DAS28-ESR > 3.2 in SSc patients. Although further studies are needed to better clarify the possible pathogenetic role of IL-21 in SSc pathogenesis and progression, the results of our study may suggest that a possible treatment directed at the neutralization of this cytokine may be useful in addition to the currently available treatment approved for joint involvement in SSc patients.

In recent years, there has been a great interest in investigating the pathophysiologic role of IL-4 in several inflammatory and autoimmune diseases [17]. IL-4 has been involved in the pathogenesis of inflammatory arthritis; however, several relevant aspects are still unclear and need to be clarified. IL-4 is secreted mainly by Th2 cells, but also by NK cells, Th1 cells, CD8^+^ T cells, innate lymphoid type 2 cells (ILC2), B lymphocytes, mast cells, macrophages, basophils, and eosinophils [35]. This cytokine is involved in B cell proliferation, collagen production by fibroblasts, and the induction of vascular cell adhesion molecule (VCAM)-1 expression in endothelial cells [35]. In a previous study, we demonstrated increased serum IL-4 levels in SSc patients compared to HCs [18]. Moreover, we showed that IL-4 was independently associated with a reduced DLco and radiological ILD in SSc patients [18]. Enhanced IL-4 concentrations were found in the synovial fluid and plasma samples of RA patients, and this was also found before disease development [17]. Increased mRNA IL-4 levels were found in the peripheral blood of RA patients and in mononuclear blood cells from RA patients after in vitro stimulation [17].

In this study, we found a positive linear correlation between serum IL-4 levels and peripheral blood CD21^low^ B cells, suggesting a possible link between this cytokine and the expansion of this B cell subset, which was also confirmed by the well-known relationship between IL-4 and IL-21. More interestingly, we showed a positive linear correlation between serum IL-4 levels and the VAS for arthritis or the DAS28-ESR, and SSc patients with moderate–high joint involvement had statistically significantly higher serum IL-4 levels than SSc patients with a DAS28-ESR ≤ 3.2. In a multivariable logistic regression analysis, we demonstrated that serum IL-4 levels were independently associated with a DAS28-ESR > 3.2 in SSc patients. We hypothesize that there is a possible role of IL-4 in the pathogenesis of inflammatory arthritis in SSc patients; however, its exact role in inflammatory arthritis has yet to be established, since the accumulating evidence implies that IL-4′s anti-inflammatory properties might be beneficial in the context of inflammatory arthritis treatment [17]. In this context, we hypothesize that IL-4 may be released to keep inflammation in check during inflammatory states.

The main limitation of our study is the lack of radiological data, which could confirm and evaluate the degree of arthritis. Moreover, there are no data about the concentrations of IL-4, IL-21, and CD21^low^ B cells in synovial fluid samples from our cohort of patients. Since, in this study, the assessment of joint inflammation was purely clinical, further studies, including a joint radiological evaluation and a synovial fluid specimen analysis, are needed to strengthen the results of this study. Finally, it would be interesting to analyze the above-reported correlations with IL-10 and TGF-beta to better understand the pathologic mechanism underlying the development of joint damage in SSc patients.

## 5. Conclusions

In conclusion, elevated peripheral blood CD21^low^ B cells and serum IL-4 and IL-21 levels were associated with higher articular disease activity in SSc patients, suggesting a possible role in the pathogenesis of SSc joint involvement.

A follow-up of these patients is required together with clinical and biological investigations to improve our understanding of the associated pathophysiological mechanisms.

## Figures and Tables

**Figure 1 jpm-13-01334-f001:**
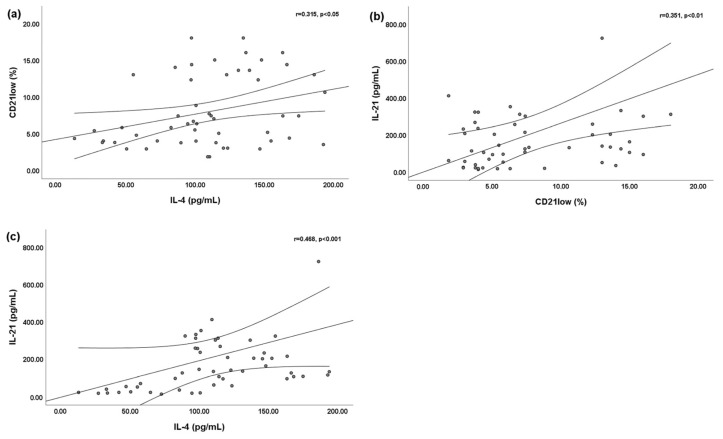
Bivariate linear correlations between peripheral blood CD21^low^ B cells and serum interleukin (IL)-4 and IL-21 levels. Circles are single values. (**a**) Positive linear correlation between peripheral blood CD21^low^ B cells and serum IL-4 level; (**b**) positive linear correlation between peripheral blood CD21^low^ B cells and serum IL-21 level; and (**c**) positive linear correlation between serum IL-4 level and serum IL-21 level.

**Figure 2 jpm-13-01334-f002:**
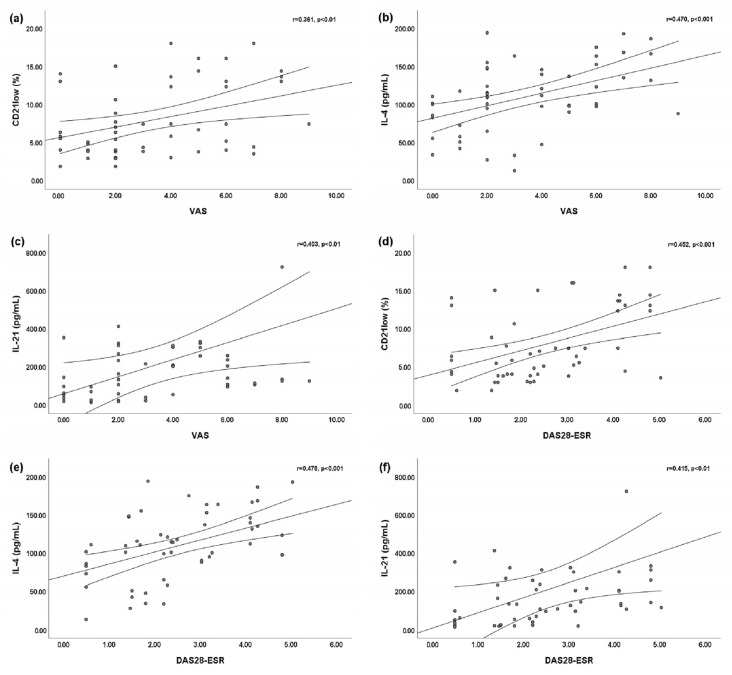
Bivariate linear correlations between the disease activity score of 28 joints based on the erythrocyte sedimentation rate (DAS28-ESR) or visual analog scale (VAS) for arthritis and peripheral blood CD21^low^ B cells, serum interleukin (IL)-4 level, and IL-21 level. Circles are single values. (**a**) Positive linear correlation between VAS and peripheral blood CD21^low^ B cells; (**b**) positive linear correlation between VAS and serum IL-4 level; (**c**) positive linear correlation between VAS and serum IL-21 level; (**d**) positive linear correlation between DAS28-ESR and peripheral blood CD21^low^ B cells; (**e**) positive linear correlation between DAS28-ESR and serum IL-4 level; and (**f**) positive linear correlation between DAS28-ESR and serum IL-21 level.

**Figure 3 jpm-13-01334-f003:**
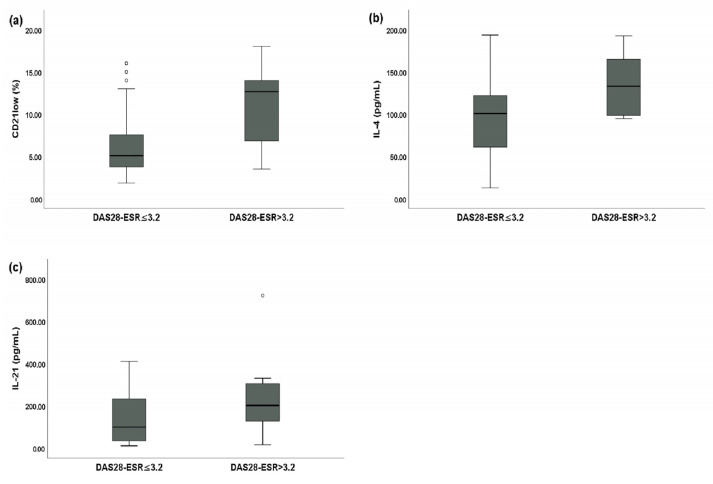
Comparative analysis between systemic sclerosis (SSc) patients with a disease activity score of 28 joints based on the erythrocyte sedimentation rate (DAS28-ESR) > 3.2 (*n* = 16) and SSc patients with a DAS28-ESR ≤ 3.2 (*n* = 36). Circles are outliers. (**a**) Median peripheral blood CD21^low^ B cell percentage for SSc patients with a DAS28-ESR > 3.2 and SSc patients with a DAS28-ESR ≤ 3.2; (**b**) median serum IL-4 levels between SSc patients with a DAS28-ESR > 3.2 and SSc patients with a DAS28-ESR ≤ 3.2; and (**c**) median serum IL-21 levels between SSc patients with a DAS28-ESR > 3.2 and SSc patients with a DAS28-ESR ≤ 3.2.

**Table 1 jpm-13-01334-t001:** Demographic and clinical characteristics of systemic sclerosis (SSc) patients (*n* = 52).

Age, Years	57.5 (48.75–63)
Female/male	48 (92.3)/4 (7.7)
dcSSc/lcSSc	23 (44.2)/29 (55.8)
Disease duration, years	11.5 (6–16)
mRSS	11 (7.75–15.25)
Anti-topoisomerase I Anti-centromere Anti-RNA polymerase III None	25 (48.1) 12 (23.1) 2 (3.8) 13 (25)
Early NVC Active NVC Late NVC	10 (19.3) 15 (28.8) 27 (51.9)
DAI	2.42 (1.26–4)
DSS	7 (6–9)
VAS for arthritis	2.5 (2–5.25)
DAS28-ESR	2.32 (1.5–3.57)
DAS28-ESR > 3.2	16 (30.77)
CD21^low^ B cells, % of total B cells	6.3 (3.97–13)
IL-4, pg/mL	110.22 (87.03–140.8)
IL-21, pg/mL	130.18 (53.93–255.02)
Previous immunomodulatory treatment MTX MMF RTX	4 (7.7) 2 (3.8) 2 (3.8)

The continuous variables are expressed as the median and interquartile range (IQR) and the categorical variables are expressed as the absolute frequency and percentage (%). dcSSc: diffuse cutaneous systemic sclerosis; lcSSc: limited cutaneous systemic sclerosis; mRSS: modified Rodnan skin score; NVC: nailfold videocapillaroscopy; DAI: disease activity index; DSS: disease severity scale; VAS: visual analog scale; DAS28-ESR: disease activity score of 28 joints based on erythrocyte sedimentation rate; MTX: methotrexate; MMF: mycophenolate mofetil; RTX: rituximab.

**Table 2 jpm-13-01334-t002:** Logistic regression analysis models showing the association between a disease activity score of 28 joints based on the erythrocyte sedimentation rate (DAS28-ESR) > 3.2 and independent variables.

		DAS28-ESR > 3.2	
		OR (95% CI)	*p*
Model 1	DAI	2.158 (1.120; 4.156)	<0.05
	DSS	0.913 (0.614; 1.357)	>0.05
	CD21^low^ B cells, %	1.301 (1.099; 1.540)	<0.01
Model 2	DAI	2.060 (1.082; 3.919)	<0.05
	DSS	0.892 (0.595; 1.340)	>0.05
	IL-4, pg/mL	1.026 (1.006; 1.045)	<0.01
Model 3	DAI	1.743 (1.022; 2.975)	<0.05
	DSS	0.985 (0.681; 1.423)	>0.05
	IL-21, pg/mL	1.006 (1.000; 1.011)	<0.05

DAS28-ESR: disease activity score of 28 joints based on the erythrocyte sedimentation rate; DAI: disease activity index; DSS: disease severity scale; OR: odds ratio; 95% CI: 95% confidence interval.

## Data Availability

The data presented in this study are available from the corresponding author upon reasonable request.

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
