# Peer review of "In Systemic Sclerosis Patients, Peripheral Blood CD21low B Cells and Serum IL-4 and IL-21 Influence Joint Involvement"

_jpm, 2023, doi:10.3390/jpm13091334_

Round 1

Reviewer 1 Report

Thank you for this interesting paper which addresses arthritis in SSc patients, an area of unmet need. 

Can the authors please add to the Table 1 the previous immunomodulatory treatments received.

Can the authors add a schematic diagram of the proposed pathways linking CD21low B cells to elevation of IL-4 and IL21 leading to joint damage.

Generally the English is clear. There are a few grammatical/translational errors that could be corrected at the stage of final editing by the Journal. 

Author Response

Reviewer #1

Thank you for this interesting paper which addresses arthritis in SSc patients, an area of unmet need.

Authors' Reply: We thank the reviewer for his/her comment.

Can the authors please add to the Table 1 the previous immunomodulatory treatments received.

Authors' Reply: We thank the reviewer for his/her suggestion to improve the quality of our manuscript and we modified table 1 accordingly.

Can the authors add a schematic diagram of the proposed pathways linking CD21low B cells to elevation of IL-4 and IL21 leading to joint damage.

Authors' Reply: We thank the reviewer for his/her suggestion to improve the quality of our manuscript and we added a schematic diagram of the proposed pathways linking elevation of IL-4, IL-21 and CD21low B cells leading to joint damage as a graphical abstract.

Reviewer 2 Report

The initial arthritis of early SSc patients is a not evident clinical symptom. The present paper well demonstrates the role of elevated level of CD21low B cells in relationship with IL-4 and IL-21. The statistical analysis of flow cytometric and immunoserologic data is appropriate and it is the main strength of the work. The interpretation and discussion of the results is relevant. It would be interesting to analyze the following correlations with IL-10 and TGF-beta to better understand the pathologic mechanism.  

Author Response

Reviewer #2

The initial arthritis of early SSc patients is a not evident clinical symptom. The present paper well demonstrates the role of elevated level of CD21low B cells in relationship with IL-4 and IL-21. The statistical analysis of flow cytometric and immunoserologic data is appropriate and it is the main strength of the work. The interpretation and discussion of the results is relevant.

Authors' Reply: We thank the reviewer for his/her comment.

It would be interesting to analyze the following correlations with IL-10 and TGF-beta to better understand the pathologic mechanism.